# Inflammation and Exosomes in Fabry Disease Pathogenesis

**DOI:** 10.3390/cells13080654

**Published:** 2024-04-09

**Authors:** Bruna Coelho-Ribeiro, Helena G. Silva, Belém Sampaio-Marques, Alexandra G. Fraga, Olga Azevedo, Jorge Pedrosa, Paula Ludovico

**Affiliations:** 1Life and Health Sciences Research Institute (ICVS), 4710-057 Braga, Portugal; a84721@alunos.uminho.pt (B.C.-R.); helenasgsilva@gmail.com (H.G.S.); mbmarques@med.uminho.pt (B.S.-M.); afraga@med.uminho.pt (A.G.F.); jpedrosa@med.uminho.pt (J.P.); 2ICVS/3B’s-PT Government Associate Laboratory, 4710-057 Braga/4805-017 Guimarães, Portugal; 3Reference Center on Lysosomal Storage Disorders, Hospital Senhora da Oliveira, 4835-044 Guimarães, Portugal; olgazevedo@yahoo.com.br

**Keywords:** Fabry Disease, inflammation, extracellular vesicles

## Abstract

Fabry Disease (FD) is one of the most prevalent lysosomal storage disorders, resulting from mutations in the *GLA* gene located on the X chromosome. This genetic mutation triggers glo-botriaosylceramide (Gb-3) buildup within lysosomes, ultimately impairing cellular functions. Given the role of lysosomes in immune cell physiology, FD has been suggested to have a profound impact on immunological responses. During the past years, research has been focusing on this topic, and pooled evidence strengthens the hypothesis that Gb-3 accumulation potentiates the production of pro-inflammatory mediators, revealing the existence of an acute inflammatory process in FD that possibly develops to a chronic state due to stimulus persistency. In parallel, extracellular vesicles (EVs) have gained attention due to their function as intercellular communicators. Considering EVs’ capacity to convey cargo from parent to distant cells, they emerge as potential inflammatory intermediaries capable of transporting cytokines and other immunomodulatory molecules. In this review, we revisit the evidence underlying the association between FD and altered immune responses and explore the potential of EVs to function as inflammatory vehicles.

## 1. Introduction

The lysosome is the main cellular recycling center, harboring, in the lumen, more than 60 different hydrolytic enzymes capable of promoting the degradation of a wide range of unwanted or unnecessary cellular components [1]. Mutations in the genes encoding for lysosomal hydrolases, membrane transporters, or accessory and trafficking proteins lead to a dysfunctional lysosomal metabolism, giving rise to disorders generally known as Lysosomal Storage Diseases (LSDs) [1,2,3].

Currently, more than 70 different types of LSD have been described, but herein, we will focus on Anderson–Fabry Disease (FD, OMIM #301500). FD was first described in 1898 by Johannes Fabry and William Anderson and presently is the second most common LSD after Gaucher Disease [4]. FD is a multisystemic disorder with an estimated prevalence of 1/2913–11,854 in newborn screenings [5,6,7,8,9]. This disorder is caused by a range of mutations in the galactosidase alpha (*GLA*) gene located in the Xq22.1 region on the long arm of the X chromosome [2,10]. As an X-linked disease, males will pass the affected gene only to their daughters while heterozygous females have half a chance to pass it to daughters and sons [10]. Females can exhibit a wide range of clinical manifestations spanning from a lifelong absence of symptoms to experiencing a phenotype as severe as the one observed in males. In contrast to the pattern seen in many other X-linked disorders, heterozygous females with FD are predominantly symptomatic. This variability in symptom presentation can be attributed to the random inactivation of one X chromosome in each somatic cell during embryonic development, a process known as lyonization [4,10].

Up to date, more than 1000 *GLA* mutations have been discovered, from which the missense type of mutation stands out due to its higher frequency [4,11]. Those mutations compromise the correct production and function of the lysosomal enzyme, α-galactosidase A (αgal-A) [2,12]. Since the αgal-A enzyme is responsible for the breakdown of complex glycosphingolipids, specifically globotriaosylceramide (Gb-3), the deficient activity of this enzyme will promote the accumulation of this molecule within lysosomes (Figure 1) [2,12].

A mutation is considered pathogenic when it causes an enzyme activity lower than 30% of the normal mean. In that case, an unnatural toxic buildup of Gb-3 will occur causing cell abnormalities and organ dysfunction, affecting particularly the cardiac, renal, and nervous systems [2,10]. Paradigmatically, two main FD phenotypes can be distinguished: the early-onset, classical severe form, if the enzymatic activity is inferior to 3% of the normal mean; and the late-onset form, in case of residual but not null enzymatic activity (>3% and ˂30%) [10].

The signs and symptoms experienced by FD patients vary according to the phenotype of the disease. Given the significant phenotypic variability among individuals sharing the same type of mutation, a clear correlation between the type of mutation and the FD phenotype is difficult to establish. Nevertheless, the specific mutation in the *GLA* gene might influence the clinical presentation of FD since some mutations are known to be associated with the early-onset phenotype and others (mostly missense mutations) with late-onset disease [10,14]. Typically, patients with the classical form go through severe clinical manifestations, starting during childhood or adolescence, including angiokeratomas, gastrointestinal symptoms, acroparesthesias, hypohidrosis, and cornea verticillata [10,14,15]. Later in life, classic FD patients experience deafness and cardiac, renal, and cerebrovascular manifestations [10,14,15]. Cardiac involvement is characterized by left-ventricular hypertrophy (LVH), heart failure, and dysrhythmias. Renal manifestations typically initiate with tubular isosthenuria and hyperfiltration, followed by microalbuminuria/proteinuria and a decline in glomerular filtration rate (GFR) that may lead to end-stage renal failure. Cerebrovascular complications englobe strokes, transient ischemic attacks, and brain white-matter lesions [14,15,16,17]. In contrast, the late-onset FD sets off in adulthood, without the early signs of the classical phenotype, and some patients may exhibit single-organ involvement, making the diagnosis of this rare disease more difficult [14,15]. It is noteworthy that males generally express more severe clinical manifestations than females, who have a wider phenotype spectrum, probably due to the lyonization [10]. For instance, LVH is observed in approximately 50% of affected males and 33% of FD females [17]. While in females, fibrosis commonly occurs without evidence of hypertrophy, in males, LVH generally precedes fibrosis and ventricular dysfunction [17]. Regarding renal complications, chronic kidney disease (CKD) is more frequent in males presenting the FD classical phenotype while proteinuria stands out as the most clinical prognostic marker for renal disease progression in both genders [17].

In Fabry Disease, the Gb-3 lysosomal deposition triggers pathogenic cascades that lead to chronic inflammation ending in cellular and organ damage [1].

In this review, cellular and molecular pathogenic mechanisms associated with immune responses in FD will be revisited. The potential of extracellular vesicles (EVs), mainly exosomes, to function as inflammatory intermediaries in FD will also be addressed. We believe that the comprehension of the immune response and the involvement of EVs in FD could greatly benefit the identification of immunological pathways that could lead to the development of targeted therapeutics and to advancements in its diagnosis and prognosis.

## 2. Cellular Mechanisms Involved in Fabry Disease Pathogenesis

Lysosomes are crucial organelles for the cellular clearance system, but they also play a central role in many other cellular processes such as energy balance mitochondrial metabolism, regulated cell death, cellular lipid trafficking, and inflammation [1,18,19]. Therefore, pathologies altering lysosome function severely disrupt cellular homeostasis [1]. In FD, the sphingolipid deposits are not sufficient to explain the pathophysiology and the intra-familial phenotypic variability. Following this thought, it has been hypothesized that Gb-3 buildup could trigger different cellular pathological mechanisms that promote the phenotypic manifestations of FD [1].

Gb-3 loading increases lysosomal pH, destabilizing the lysosome, which leads to a dysfunctional autophagy lysosome pathway (ALP) [18]. ALP is a highly regulated process that can be divided into three main sub-systems: macroautophagy, microautophagy, and chaperone-mediated autophagy. Despite the different biochemical and cellular pathways used by each sub-system, all three share the same purpose of degrading and recycling exhausted cytosolic proteins and organelles within cells [20]. Macroautophagy is the main pathway of ALP; hence, hereafter, it will be termed autophagy. Autophagy can be divided into four main steps: initiation, elongation, closure, and fusion. Briefly, it starts with phagophore formation that progressively matures into an autophagosome (double-lipidic membraned vesicle). Eventually the autophagosome fuses with the lysosome, generating an autolysosome, where the autophagosome’s cargo is degraded by acid hydrolases (Figure 2) [20].

Although the role of autophagy in FD is not entirely understood, its impairment has already been reported in kidney cells, fibroblasts, and lymphoblasts obtained from FD patients [22]. The same study suggests a contribution of the disrupted autophagy to the renal pathology experienced by many FD patients. This raises the hypothesis that the impairment of the autophagic pathway may promote some of the pathological events of FD [22]. Corroborating the disruption of autophagy associated with FD, a statistically significant increment in the mean expression of LC3 was observed in the conjunctival epithelium in FD patients [23] and in the gray matter of the brains of αGal-A-deficient mice [24]. A decreased ratio of LC3-PE to LC3 I, BECLIN1, and autophagosomes was also reported in the kidneys of FD patients, further suggesting decreased autophagy flux [25]. An accumulation of autophagic vacuoles and intense signal of p62/SQSTM1 and ubiquitin were also observed in the renal biopsies of FD patients [22].

The gathered evidence indicates that in FD, autophagy dysfunction might consequently activate several signaling pathways leading to hypertrophy, apoptosis, necrosis, and fibrosis [26]. Furthermore, it has also been described that the intracellular accumulation of Gb-3 leads to the oxidative damage of proteins and DNA, culminating in cell death [27]. Together, data have shown that cells from FD patients present increased levels of apoptosis, revealed by the incrementation of apoptotic markers such as mitochondrial membrane depolarization, in comparison to non-Fabry control cells [28]. Congruently, reduced levels of apoptosis have been observed in cells from FD patients under enzyme replacement therapy (ERT) compared to untreated ones [28]. Importantly, disrupted autophagy and higher levels of apoptosis are inherent to all LSDs, regardless of the accumulated substrate [28,29].

As previously mentioned, lysosomes are central players for normal inflammatory responses and immune system function since they are involved in processes such as antigen presentation, phagocytosis, the secretion of perforins by cytotoxic T cells, and the release of pro-inflammatory factors among others [1,2]. Studies have demonstrated that substrate buildup in lysosomes triggers several cascades that result in persistent inflammatory responses, increasing cellular damage, potentially becoming pathogenic, independent of the substrate involved [1].

Globotriaosylsphingosine (lyso-Gb3, the deacylated form of Gb-3) was shown to be linked to FD pathophysiology by inducing autophagy in ARPE-19 cells (a retinal-pigment epithelial cell line) [30]. Also, it was observed that 3-MA (an autophagy inhibitor) alleviated lyso-Gb3-induced necroptosis [30]. These results suggest that lyso-Gb3 induces retinal-pigment epithelial-cell-derived endothelial dysfunction via an autophagy-dependent necroptosis pathway, triggering inflammation [30]. Importantly, the same authors showed that conditioned media of lyso-Gb3-treated ARPE-19 elicited endothelial necroptosis, inflammation, and senescence via the autophagy-dependent necroptosis pathways in human umbilical-vein endothelial cells (HUVECs) [30]. In such ways, an interplay between the autophagic, necrotic/apoptotic, and inflammatory pathways seems to establish a driving force for FD pathology.

Importantly, Gb-3 loading has been also demonstrated to promote the dysfunction of several key endothelial pathways, which may prompt vasculopathy, a driving force for the life-threatening complications of FD [31,32,33]. In a study, cells from a vascular endothelial cell line exposed to Gb-3 overexpressed several CAMs including ICAM-1, VCAM-1, and E-selectin [34]. The overexpression of CAMs triggered by Gb-3 may be mediated by reactive oxygen species (ROS), considering their known participation in the upregulation of endothelial CAMs through the activation of transcription factors, particularly NF-κB [34]. The incrementation of CAMs leads to higher levels of the vasoconstrictor cyclooxygenase-2 (COX-2) and facilitates the internalization of calcium-activated potassium channels. In turn, a downregulation of intracellular calcium levels and endothelial nitric oxide synthase (eNOS) occurs [35,36,37]. Following *GLA* knockdown, hybrid endothelial cells demonstrated a remarkable decrease of over 60% in eNOS activity [38]. In agreement, the presence of Gb-3 in human microvascular cardiac endothelial cells inhibited total eNOS expression in both unstimulated and stimulated cells (by TNF-α) and yet also promoted inducible nitric oxide synthase (iNOS) expression, and augmented the expression of COX-2, in unstimulated cells [35]. The latter decreases nitric oxide bioavailability and increases the formation of ROS, promoting oxidative stress [35,39]. Untreated classical FD male patients revealed over six-fold higher levels of serum 3-Nitrotyrosine (3-NT), a biomarker of oxidative damage, than aged- and gender-matched controls [38,40]. Moreover, Gb-3 buildup was positively linked to angiotensin I and II production in the vascular endothelium [35,39,41]. Collectively, these lead to the prothrombotic, vasoconstrictive, and pro-inflammatory phenotype that describes FD patients.

Altogether, the data have shown that both the innate and the adaptive immune systems are clearly operating in FD patients. Yet, how they are activated, or which pathways are triggered, is still not fully understood. Hereafter, this topic will be further explored in detail.

## 3. Role of Inflammation in Fabry Disease: A Friend or a Foe?

The purpose of immune responses is to ensure that the host adapts to abnormal conditions, always aiming for the restoration of the functionality and homeostasis of organs and tissues [42,43,44]. Succinctly, the immune system consists of two main branches: the innate and the adaptive [42,43]. Following the classic dichotomy, while the innate system stimulates a non-specific inflammatory reaction, the adaptive response elicits a highly specific response to antigens, developing long-term memory [1,43]. As such, different cell populations participate in these two branches of host immunity. The innate immune response involves cells such as neutrophils, mast cells, mononuclear phagocyte cells (monocytes, dendritic cells (DCs), and macrophages), and some types of lymphocytes (gamma delta T cells, natural killer cells, and innate lymphoid cells). On the other hand, B and T lymphocytes belong to the adaptive immune system [43]. Note that a suitable and successful immune response requires a regulated and organized crosstalk between the two immune system branches, which is accomplished by soluble factors and cell-to-cell interactions [43]. In the past few years, an increasing body of evidence has been defying the dogma of the innate immune system, being characterized by a non-specific and non-immunological memory response [45]. A role of the innate immune system in reinfection or cross-protection (from a different microorganism that first originated the infection) has been proposed, being termed “trained immunity” [45]. Nonetheless, it is necessary to underline that the definition of memory used in the context of the innate immune system is different from that used for the classic adaptive immune response, which demands a high degree of specificity and amplification [45]. Acute inflammation is an example of a response involving the immune system that is pivotal for the immediate protection of the organism against multiple pathogens, noxious stimuli, or physical injury [42,44,46]. Briefly, the inflammatory response requires the coordinated supply of blood elements, such as plasma and leukocytes, to the site of injury or infection [44]. It is initiated by resident immune cells, particularly macrophages and mast cells, through pattern recognition receptors (PRRs) like Toll-like receptors (TLRs) [43,44,47,48,49].

FD is characterized by a chronic low-grade systemic inflammation, which apparently results from the interaction between Gb-3 and TLRs [1]. Gb-3 accumulation may behave as a damage-associated molecular pattern (DAMP) or cause DAMP production by injured cells. The accumulation of Gb-3 can induce the secretion of cellular adhesion molecules (CAMs) and cytokines, increase the production of an extracellular matrix (ECM), and enhance an accumulation of ROS, collectively promoting a pro-inflammatory effect on leukocytes and endothelial cells [1,34]. Over time, this chronic inflammatory state will cause the development of fibrosis in different tissues, mainly in the heart and kidney, thereby resulting in associated clinical manifestations [50].

Hereafter, we will address this topic by separately analyzing the two main pillars of the inflammatory process: alterations observed in soluble factors and their communicating receptors in FD and the immune cell profiles of FD patients. A brief assessment of the impact of FD treatment on the immune response will also be described.

### 3.1. Inflammatory Soluble Factors: Profiles in Fabry Disease

Several studies have addressed the pro-inflammatory cytokine profiles and the subsets of innate immune cells responsible for cytokine secretion in FD patients [1]. However, the generated data are still controversial and sometimes lack elucidation regarding the therapeutic regimens (and their initiation periods) that FD patients are on, which may influence both the observed results as well as their interpretation.

DCs and freshly isolated peripheral blood mononuclear cells (PBMCs) from FD patients showed the production of higher levels of IL-1β and TNF-α compared to normal controls whereas the supernatants of 24 h PBMC cultures revealed significantly increased levels of IL-6 and IL-1β [51] (Figure 3). In DCs and monocytes from FD patients, the accumulation of Gb-3 prompted the release of IL-1β and TNF-α, but not IL-6 [51]. Moreover, Rosa Neto et al. observed that the serum levels of IL-6 and TNF-α were significantly higher in FD patients [52]. Yet, there was no difference in the levels of IL-1β between FD patients and healthy individuals [52] (Figure 3). TNF-α and IL-1β are hallmarks of autoinflammatory diseases, disorders of the innate immunity with a preponderance of neutrophil and monocyte/macrophage-driven inflammation and inappropriate cytokine-mediated pathology [1,47,51]. The augmented secretion of IL-1β is probably a consequence of the recognition of an abnormal self or danger signals from injured cells. To this extent, FD reveals characteristics similar to those of autoinflammatory disorders, whereby Gb-3 loading may be detected as a danger signal [51]. Higher plasma protein levels of fibroblast growth factor 2 (FGF2) and IL-7 were also observed in FD patients compared to healthy controls [53] (Figure 3). FGF2 regulates angiogenesis, cell growth, and tissue repair by binding to the fibroblast growth factor receptor (FGFR). Thus, FGF2 upregulation may cause endothelial impairment. IL-7 is a cytokine required for T-cell survival, and is involved in autophagy regulation, so its increased levels in FD may indicate an attempt to induce autophagy, but increased IL-7 levels also lead to inflammatory dysfunction [53]. In agreement with other studies, in FD patients’ plasma, significantly elevated levels of inflammatory markers, namely IL-6, TNF-α, metalloproteinase (MMP)-2, MMP-9, and galectin-1, were found [54] (Figure 3). The incrementation of these inflammatory markers clearly suggests a systemic and chronic inflammatory state in FD patients [54].

In FD patients, the secretion of inflammatory mediators, such as TNF-α and IL-1β, seems to be gender-dependent. Üçeyler and colleagues detected higher gene expression levels of TNF-α and IL-1β in naïve PBMCs of male FD individuals. Concerning females, no differences were observed between FD and control groups [55] (Figure 3). The upregulation of TNF-α transcripts was found to be specifically associated with males suffering from pain rather than those without pain. However, an increase in TNF-α secretion from PBMCs in males with FD was observed irrespective of their pain status when compared to healthy control individuals [55]. An analysis of FD patients based on gender revealed that females FD patients presented significantly higher serum levels of TNF-α than the healthy female controls [52]. FD male patients had significantly increased serum levels of IL-6 and TNF-α compared to control males [52].

Concerning FD manifestations, patients with LVH also had greater plasma levels of the inflammatory mediators, like TNF, IL-6, and MMP-2, and patients with late gadolinium enhancement (marker of myocardial fibrosis) revealed higher plasma levels of MMP-2, mid-regional pro-atrial natriuretic peptide (MR-proANP), and B-type natriuretic peptide (BNP) [56] (Figure 3). Furthermore, FD patients with diastolic dysfunction presented raised amounts of plasma MR-proANP, BNP, and MMP-2. At last, patients with renal complications (patients with mildly or moderately reduced renal function and estimated GFR < 60 mL/(min·1.73 m^2^)) showed increased plasma levels of BNP, MR-proANP, TNF-α, MMP-2, MMP-8, galectin-3, and galectin-1 [54].

Focusing on the crosstalk between soluble factors and signaling receptors, an antagonistic expression of CD1d at lower levels and MHC II being overexpressed on the plasma membranes of monocytes in FD patients were observed [57] (Figure 3). Additionally, FD patients showed higher levels of TNF receptor 1 (TNFR1) and TNFR2 (two different receptor subtypes of TNF-α) [54]. The same study revealed increased levels of TNFR1 and TNFR2 in FD patients with renal complications [54]. FD patients with LVH had higher plasma levels of TNFR2 while patients with late gadolinium enhancement presented increased levels of TNFR1 and TNFR2 [56]. Also, evidence showed that the accumulation of Gb-3 underlies the activation of receptors such as Toll-like receptor 4 (TLR4), which may initiate Notch1 signaling and, in turn, activate NF-κB [1]. As a result, pro-inflammatory cytokines are produced, inducing a local and systemic inflammatory process [1]. Pre-treating monocyte-derived macrophages from healthy individuals with a TLR4-blocking antibody, prior to the supplementation with Gb-3 and DGJ, prevented the production of TNF-α and IL-1β in a study [51]. These results support the hypothesis that the blockading of TLR4 prevents the effects promoted by Gb-3 accumulation, which indicates Gb-3 as a ligand for TLR4 [47,52].

Fibrosis, driven by TGF-β1 and TLR4 stimulation, is a common factor in the kidney and heart in FD patients and a trait analogous with diabetic nephropathy [1,58,59]. Therefore, Gb-3 appears to stimulate TLR4 in other cell types, such as podocytes, in addition to immune cells. Notably, lyso-Gb3 can be released from cells, reach circulation, and enhance TGF-β1 production through Notch1 activation [58,59].

### 3.2. Inflammatory-Cell Profile in Fabry Disease

The pro-inflammatory profile observed in FD patients is also associated with alterations in inflammatory cells. Accumulating evidence shows that FD patients have a reduced number of monocytes, CD8^+^ T cells, and DCs and increased percentages of total lymphocytes [57,60]. The decreased numbers of monocytes and DCs are potentially associated with a reduced production of immune cells in bone marrow and with an incrementation of the apoptotic rate and/or their increased migration to peripheral tissues [57]. Although no differences were observed in the numbers of dorsal root ganglia (DRG) macrophages and T cells in old *GLA* knockdown mice, lower numbers of anti-inflammatory M2 macrophages were detected, suggesting a reduced anti-inflammatory polarization of macrophages [61] (Figure 3).

Given the previously mentioned lower expression of CD1d in FD patients and considering that CD1d mediates the invariant natural killer T (iNKT) cell activation, this cell population might also be deregulated [57]. iNKT cells represent a subset of T cells that can be grouped into three main different groups: positive only for CD8 (CD8+ iNKT), positive only for CD4 (CD4+ iNKT), and negative for both molecules (DN iNKT). Hence, the three subgroups express a differential cytokine phenotype [57,62] (Figure 3). Consistently, iNKT cells from αgal-A-deficient mice showed signs of persistent activation, suggesting that the iNKT-cell antigen might be an αgal-A substrate [63]. Furthermore, studies performed both in FD patients and in a mouse model showed a reduction in the number of CD4+ iNKT cells and an increase in the number of DN iNKT cells [1,62]. Remarkably, a reduction in IL-4 levels was also observed, probably due to the decrease in both CD4+ and CD4− iNKT cell counts, but essentially in the number of CD4+ iNKT cells, since this is the subset responsible for most of the IL-4 produced (Figure 3). Nonetheless, no alterations were found in the interferon (IFN)-γ production. Contradictory results have been found with respect to the proportion of CD8+ iNKT cells [1,62]. While Rozenfeld and collaborators noticed a higher proportion of CD8+ iNKT cells in samples from control individuals when compared to FD patients (26% vs. 19%, *p* < 0.01) [1,62], Pereira et al. did not observe significant differences [1,62]. Altogether, these findings suggest a predisposition to a T helper type 1 (Th1) immune profile in FD iNKT cells, exposing a pro-inflammatory environment [62]. Furthermore, as discussed above, Gb-3 deposits in FD seem to be the trigger of an innate immune response related to CD1d and TLR4 pathways.

### 3.3. Influence of FD Treatment on Inflammatory Response

Despite FD being a disorder without cure, there are treatments that can improve FD patients’ quality of life and protect organ dysfunction, increasing survival. Nowadays, the cornerstone of FD treatment is ERT with agalsidase-alfa or -beta, a lifelong therapy [64]. Such treatments focus on the replacement of the αgal-A enzyme to avoid the Gb-3 and lyso-Gb3 buildup in cells throughout the body [64]. Oral chaperone therapy, involving a molecule that alters the defective forms of the αgal-A enzyme, transforming them into functional structures, is also an option, enhancing residual enzyme activity in patients with amenable mutations [64]. To our current knowledge, there have been no studies exploring the effect of chaperone therapy in FD immune response. Concerning ERT, the literature reports controversial findings. For instance, De Francesco et al. reported no statistical differences in the expression or production of TNF-α, IL-1β, IL-6, IL-13, and INF-γ between FD patients undergoing ERT and FD patients receiving no treatment [51]. In contrast, Yogasundaram et al. observed that ERT-treated FD patients revealed higher levels of TNF, TNFR1, TNFR2, MMP-2, and lyso-Gb3 compared to those not under ERT [54] (Figure 3). These findings highlight the fact that patients who qualify for and receive ERT typically have more severe disease manifestations, which may limit ERT efficiency. These results suggest that although plasma lyso-Gb3 is a potential primary screening biomarker for classical and late-onset FD [65], cautions should be taken when analyzing it in FD patients under ERT. In addition, higher serum levels of IL-6 and TNF-α but not of serum IL-1β were reported in patients under ERT in relation to patients not treated with ERT [52]. Fu et al. showed that MCP-1 and TNF-α concentrations were almost two-fold higher in plasma from ERT-treated FD patients, which suggests that ERT is not able to reduce the established systemic inflammatory state [66]. Also, the chemokine receptor CCR2 was double-expressed on monocytes from patients with low estimated GFRs [66] (Figure 3). Concurrently, the researchers observed an increase in the number of IL-12b transcripts (encoding IL-12p40, a subunit of IL-12) in unstimulated PBMCs over a 2-year period of continuous ERT [66]. An increased GM-CSF count in plasma from ERT-treated FD patients compared with healthy controls was also reported [66] (Figure 3). The chemokine MCP-1 interacts with its receptor CCR2, promoting monocyte infiltration into injured tissues [66]. Moreover, strong activations of the complement system, evidenced by high C3a and C5a serum levels, in male FD patients before and after ERT were reported [67]. Contrary to the decrease in lyso-Gb3 levels under ERT, C3a and C5a levels markedly increased in FD patients with nonsense mutations under ERT. These FD patients presented anti-drug antibodies (ADA) whereas FD patients with missense mutations, which were ADA-negative, showed heterogenous C3a and C5a serum levels under ERT [67]. In addition, a prominent increase in IL-6, IL-10, and TGF-b1 serum levels was observed in FD patients with missense mutations under ERT, most of whom developed mild nephropathy with decreased estimated GFRs [67]. Therefore, a strong complement activation in FD independently of ERT therapy, especially in males with nonsense mutations and the development of ADAs, was later proposed [67]. The chronic inflammation was demonstrated to be a driver of organ damage in FD that seems to progress despite ERT [67]. In contrast, in a study, ERT was shown to significantly reduce the levels of lyso-Gb3 and IL-6, IL-2, IL-1β, TNF-α, and MCP-1 [68]. Such alterations were positively correlated with changes in the left-ventricular mass index (LVMI) that had also been prompted by ERT [68].

The non-consensual findings aforesaid have demonstrated the long road that still needs to be covered in this topic. Some evidence suggests that dysregulated inflammation persists or is even worsened in patients under ERT while other studies indicate an improvement in the inflammatory response. This raises some hypotheses: (a) ERT fails to adequately reset the Gb-3 and lyso-Gb3 levels and, consequently, fails to suppress the inflammatory process; (b) ERT itself contributes to the inflammatory process, corroborated by the development of ADAs; and (c) there is a point of no return after which ERT is no longer beneficial for a specific target organ, with FD progressing and, eventually, leading to organ failure despite the treatment [52,54]. In fact, the literature has been questioning the benefit of ERT, especially in patients already presenting organ damage [69,70,71]. Its efficiency seems to depend on different factors, among which are phenotype, patient gender, initiation age of ERT, dose of ERT, type of ERT (agalsidase-alfa or -beta), and the generation of ADAs [72,73,74,75,76]. Conducting research on this topic would be fundamental since monitoring patients after the initiation of ERT using inflammatory biomarkers would provide valuable information about disease control and long-term prognosis [54]. In conclusion, accumulating evidence suggests that identifying differentially expressed inflammatory proteins in FD patients holds promise as both diagnostic and prognostic biomarkers. Such markers not only aid in discriminating the inflammatory processes amplified by the disease but also offer insights into the pathophysiological mechanisms driving organ damage [1].

## 4. Extracellular Vesicles in Fabry Disease

In the past, the cells’ ability to release membrane vesicles to the extracellular environment had only been identified during apoptosis and as a mechanism of waste management. Currently, this capacity is also recognized in healthy cells as a cell-to-cell communication mechanism involved in physiological and pathological processes [77]. The generic terminology “extracellular vesicles” (EVs) is used to refer to all type of vesicles secreted in either healthy or pathological conditions. EVs are defined as a heterogeneous group of lipid bilayer membrane-limited vesicles deriving from the endosome or plasma membrane that can broadly be categorized into three groups: exosomes, microvesicles (MVs), and apoptotic bodies [77] (Figure 4). These three subsets of EVs can be distinguished according to their biogenesis (cellular origin), function, content, and size (Table 1) [77].

Exosomes and MVs have been identified as mediators in intercellular communication, which has made their study imperative since intercellular communication is vital to the homeostasis of biological systems [78]. Hereupon, exosomes and MVs can convey information from their parent cell to a target cell either through the secretion of their cargo, such as small soluble molecules, or through EV–cell contact [77]. Of note, EVs’ communication can target neighboring cells or distant ones, with this process being highly dependent on the recipient cell’s surface receptors [77,79].

Considering the demonstrated activity of EVs as signaling vehicles sharing combinatorial information between cells throughout the body, these vesicles present the potential to mediate the inflammatory process observed in FD patients [77]. Among the different types of EVs, exosomes have already been proven to influence a wide range of pathological conditions such as cancer, neurodegenerative and liver disorders, and heart failure among others [80]. The latter is possibly due to the EVs’ cargo and surface receptors that enable them to play a wide spectrum of functions in diverse processes including cellular migration and in metastatic disease, the regulation of gene transcription and translation, the balancing of immune response, and the regulation of central and peripheral immunity, apoptosis, and angiogenesis and wound healing [80].

### 4.1. Exosomes

Exosomes are lipid bilayer-membrane vesicles derived from intraluminal vesicles (ILVs) found within multivesicular endolysosomal compartments that, upon fusion with the plasma membrane, are excreted to the extracellular environment [81].

Exosomes are found in various biological fluids (e.g., urine, blood, and saliva) and are produced by many types of immune cells including DCs, T and B lymphocytes, macrophages, mast cells, and reticulocytes, as well as other types of cells like tumor cells and active neurons [79,82]. Exosomes are characterized by their surface receptors and cargo (proteins, lipids, and nucleic acids [83]), revealing pivotal roles in their mobility and uptake and in their capacity to transfer messages from one cell to another, respectively [80]. During biogenesis, exosomes engulf cytosolic parent-cell components, with the exosomal content being a mirror of the physiological or pathological state of the origin cell. Therefore, an exosomal-cell-type signature can be drawn that may have an extreme significance in the diagnosis and management of different diseases [77,83,84,85]. Nonetheless, a conserved set of components has been recognized in exosomes’ cargo, being extremely well defined by the endosomal formation of exosomes [77,83,84] (Figure 4). To illustrate, the Ras superfamily of monomeric G proteins (Rab); apoptosis proteins; adhesion molecules; heat shock proteins (HSC73 and HSC90); tetraspanins (CD9, CD81, CD82, and CD63); annexins I, II, V and VI; cytoskeletal proteins (actin and albumin); and GTPases are examples of proteins commonly found in exosomes [83]. On the other hand, there are parent-cell-specific proteins such as major histocompatibility complex (MHC) II and CD86 from antigen-presenting cells (APCs); HP60 from cardiomyocytes; and granzymes and perforins from cytotoxic T cells and platelets, respectively [83].

Concerning lipids, exosomes have increased ratios (up to four times greater) of these macromolecules in relation to their origin cells, which may be responsible for the enhanced rigidity of the exosomal membrane. Examples of lipids comprising the exosomes’ cargo are cholesterol, sphingolipids (ceramide and sphingomyelin), glycerophospholipids, prostaglandins, and leukotrienes [83].

In addition to proteins and lipids, exosomes’ cargo can also include nucleic acids, which are important signaling molecules that, due to their potential interference with transcription and translation machineries of the target cell, are the focus of extensive study. Those include DNA, mRNA, transfer RNA (tRNA), ncRNA (such as miRNA, long non-coding RNA (lncRNA), and small cajal body-specific RNA (scaRNA), ribosomal RNA (rRNA), small nucleolar RNA (snoRNA), small nuclear RNA (snRNA), and piwi-interacting RNA (piRNA)) [86,87].

In a study, in FD patients with stable renal function (SRF) and progressive nephropathy, miR-21-5p and miR-222-3p expressions were upregulated in urinary extracellular vesicles (uEVs), suggesting that these miRNAs may play a crucial role in the initial phases of the disease progression [88]. Conversely, miR-30a-5p, miR-10b-5p, and miR-204-5p exhibited downregulation in patients with progressive nephropathy [88]. These differentially expressed miRNAs derived from uEVs were found to be linked with signaling pathways that are recognized as crucial in the development and progression of nephropathy in various other diseases [88]. Moreover, FD patients were also characterized by higher levels of miR-126-3p in plasma EVs that are augmented with age [89]. In vitro, the glycosphingolipid buildup was shown to lead to premature senescence and increased miR-126-3p levels [89].

Moreover, a significantly increased exosomal secretion was observed in a FD cell model [90] while Rab GTPases (involved in exocytotic vesicle) were demonstrated to be significantly downregulated in *GLA*-knockout human embryonic stem cells (hESCs) [81]. In the former study and after ERT (agalsidase-beta), the exosome secretion significantly decreased [90]. Such findings were interpreted as representing an attempt by FD cells to avoid the excessive and cytotoxic loading of Gb-3 through intracellular pathways, including increased p53 expression [90].

In such a manner, exosomes, due to their cargo and surface receptors, can play a range of functions on diverse processes such as the balancing of immune responses and the regulation of central and peripheral immunity, apoptosis, angiogenesis and wound healing, and host–microbiome interaction and viral immunity [80]. Therefore, exosomes may have an important role in the progression of FD through the modulation of the inflammatory cascades known to occur in this disorder.

### 4.2. Exosomes as Inflammatory Mediators

Exosomes were first considered to be involved in immunologic responses when investigators noticed that they were secreted by immune cells harboring an exosomal cargo and set of surface receptors with relevant roles in triggering the immunologic cascade such as MHCI, MHCII, and T-cell co-stimulatory molecules [91]. This first piece of evidence was strengthened by the discovery that exosomes derived from non-immune cells also played a key role in immune reactions since they expressed MHCI on their surfaces and carried immunomodulatory molecules [91].

The available literature indicates that various pro- and anti-inflammatory cytokines, including IL-2, -4, -10, -18, and -33; TNF-α; TGF-β; and macrophage-colony-stimulating factor (M-CSF), are preferentially enriched within exosomes [84,92,93] (Figure 4). Cytokines could also be carried as exosomes’ membrane proteins, named as exosome-associated cytokines (EACs), or transported inside exosomes [78,94] and protected from environmental degradation [93]. Hence, exosomes loaded with cytokines become biologically active upon interaction with target cells such as macrophages or cells expressing specific cytokine receptors [95,96]. For instance, exosomes secreted by monocytes and packed with IL-1β can stimulate endothelial cells and trigger the production of IL-1β by other monocytes in an autocrine way, revealing, then, a pro-inflammatory action [97] (Figure 4). Exosomes can also function as mediators of the immune and inflammatory pathways via their lipid fraction since this content is capable of activating TLR4 in macrophages [96].

The idea that exosomes can interfere with the immune cascade, consequently, promoting the progression or remission of disease, has been widely explored in the cancer field. In this context, Pagetet, in their “seed in soil” model, first suggested that the formation of metastatic tumors implies the creation of a pre-metastatic niche in distal sites [98], later shown to be due to the capability of tumor-derived exosomes to transport cytokines and growth factors to cells at distant localizations [86]. In addition, Costa-Silva and collaborators reinforced this idea, with the observation that pancreatic ductal adenocarcinoma (PDAC)-derived exosomes initiated a pre-metastatic niche in the liver by fostering a pro-inflammatory environment appropriate for metastasis [99]. Furthermore, De Wever and colleagues studied the part of tumor-derived exosomes packed with TGF-β and noticed that these exosomes promote vascularization, tumor growth, and local invasion through the conversion of fibroblasts into myofibroblasts [100].

Tumor-derived exosomes can also impact immune cells, leading to the suppression of immune responses. For instance, they can potentiate the apoptosis of activated T cells, suppressing the generation and expansion of regulatory T cells (Tregs) and inhibiting DC maturation, due to their content of Fas ligand (FasL), tumor necrosis factor apoptosis-inducing ligand (TRAIL), and TGF-β among other molecules [78]. Liu et al. also demonstrated that tumor-derived exosomes prevent the activity of NK cells by compromising their cytolytic activity and suppressing their proliferation [101]. Still regarding tumor-derived exosomes, breast cancer cells were found to produce exosomes that lead to the activation of tumor-associated macrophages (TAMs), resulting in the secretion of pro-inflammatory cytokines [101]. Additionally, EVs from mesenchymal stem cells can impede the immune response by transferring miRNAs (e.g., miR-451a, miR-1202, miR-630, and miR-638) that target the TLR signaling and the NF-kB pathways in macrophages [12].

In summary, exosomes packed with cytokines or with cytokines associated with their membranes are able to modulate the host immune system by triggering B and T cells, promoting or preventing cytokine production in other cells, activating or downregulating inflammatory pathways in target cells, and stimulating the migration of granulocytes to inflamed tissues [92]. In this way, exosomes might be involved in the inflammatory and immune pathways of FD by transporting specific cargo protected from degradation to a distant target organ. This might contribute to FD progression, namely towards a predominant renal or cardiac involvement. Given the proven potential of exosomes to play roles as biomarkers in the diagnosis and prognosis of several diseases [92], the utility of these vesicles may be further expanded to FD diagnosis, prognosis, and early detection with a cautionary note for their use in heterozygous females given their heterogenous disease phenotype. Exosomes have already expanded horizons in terms of FD management. In 2020, the first steps were taken by studying the usage of engineered EVs as vehicles for GLA delivery [102]. In vitro and in vivo studies showed that EVs carrying GLA (EV-GLA) were rapidly up-taken and driven directly to the lysosomes, restoring enzyme functionality much more efficiently than clinical-reference agalsidase-alfa [102]. In vivo, EVs were also well tolerated and distributed among all main organs including the brain. This represents a crucial advantage of the usage of EVs as therapeutic protein delivery systems over ERT, which fails to access the brain [103]. In addition, a single intravenous administration of EV-GLA was able to decrease Gb-3 levels in clinically relevant tissues, namely kidneys and brains [103].

Altogether, and considering the potential of exosomes as intermediators of inflammatory responses in several disorders, the utility of these EVs in the management of FD patients should be further explored, bringing new perspectives to diagnosis, prognosis, and early detection.

## 5. Conclusions

The accumulated evidence clearly demonstrates the involvement of inflammation in FD despite the lack of knowledge concerning the exact cellular and molecular mechanisms connecting the intracellular accumulation of glycolipids with the inflammatory process and associated organ pathology [1,2]. Nonetheless, strong evidence suggests that Gb-3 buildup functions as a stimulator of innate immune mechanisms through the activation of TLR4 and CD1d pathways, thus triggering inflammatory responses via immune cells such as iNKT cells [34,49]. The continuous exposure to Gb-3 observed in FD patients potentiates the progression of acute to chronic inflammation, contributing to the permanent release of pro-inflammatory mediators, which eventually leads to organ damage, enhancing the pathologic cycle of the disorder [47,49]. Despite that, it is still not completely understood how FD triggers the mechanisms underlying immune activation and, in return, how those prompted immune mechanisms promote pathological changes in FD target organs, particularly the kidney and heart. As follows, further research exploring phenotype-specific inflammatory profiles is needed to shed light on the mechanisms underlying the inflammatory process occurring in FD. In this way, new advances in targeted anti-inflammatory therapies could emerge, reflecting a potential to revolutionize the management of FD.

Regarding EVs, their involvement in the immune and inflammatory pathways is unquestionable, but how exactly can EVs affect the progression of FD? Might EVs be responsible for the fate of the disease? What is the role of EVs regarding intercellular communication? Is it a protective or a harmful role? Can the immunological information transported by EVs influence other cellular mechanisms, such as autophagy and apoptosis, in FD? Might EVs represent a new source of biomarkers for FD diagnosis and progression? These are questions that remain to be addressed but whose answer could revolutionize the early diagnosis of target-organ damage in FD, as well as its management, by taking advantage of the particularities of exosomes to transport cargo in a targeted manner without the consequences of degradation. For example, would it be possible to modulate exosomes activity in order to improve the efficacy of the current treatments or to develop a new therapeutic tool of FD?

## Figures and Tables

**Figure 1 cells-13-00654-f001:**
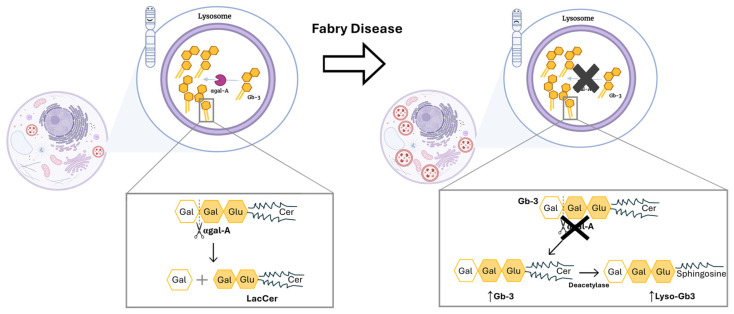
Mutations in the GLA gene alter the structure of αgal-A and lead to its markedly deficient or absent activity. **Left side**: Functional αgal-A activity in a person with no mutation in *GLA* gene. As a result, there is no accumulation of unwanted substrate in lysosome. Also, the enzyme is responsible for removing the terminal galactose sugar from Gb-3, which leads to the formation of lactosylceramide (LacCer). **Right side**: The function of αgal-A is altered due to the presence of a mutation in *GLA* gene. The latter leads to a buildup of Gb-3. Thus, an accumulation of dysfunctional lysosomes occurs within the cell. Cer: ceramide; Gal: galactose; Gb-3: globotriaosylceramide; Glu: glucose; LacCer: lactosylceramide; Lyso-Gb3: globotriaosylsphingosine. Image has been adapted from [13].

**Figure 2 cells-13-00654-f002:**
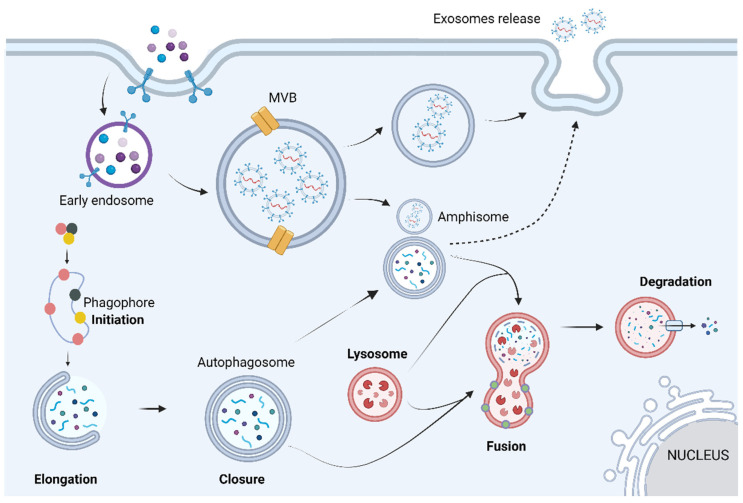
The autophagic process can be divided into mechanistically different phases: initiation, vesicle formation (elongation and closure), autophagosome–lysosome fusion, and breakdown of the cargo. Briefly, these steps are based on the appearance of an isolation membrane in the cytoplasm, known as phagophore, that engulfs a portion of the cytoplasm, expands, and seals itself to form the autophagosome, a double-membrane structure. The autophagosome may also fuse with endosomes or multivesicular bodies, forming a structure known as amphisome. At a final stage, the outer membrane of the autophagosome fuses with the lysosome, giving origin to an autolysosome, in which all the autophagosome content is degraded by acidic lysosomal hydrolases. Image has been adapted from [21]. Image: created with biorender.com.

**Figure 3 cells-13-00654-f003:**
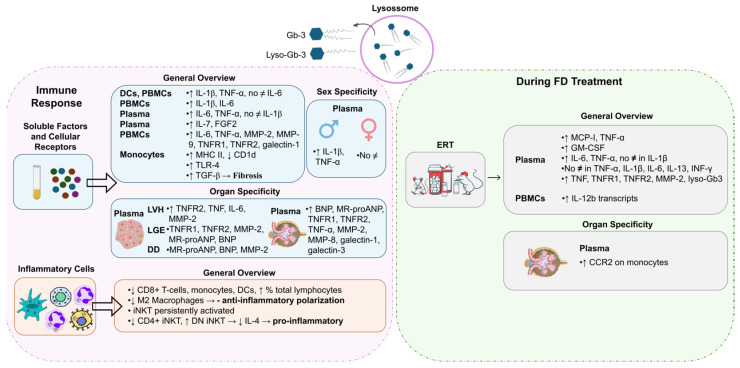
Activation of the immune response in Fabry Disease (FD). Globotriaosylceramide (Gb-3) and globotriaosylsphingosine (Lyso-Gb3) influence the levels of soluble factors and signaling receptors and inflammatory-cell profile during FD. Different immune responses may be triggered in FD patients with different genders or distinct disease manifestations. FD treatment, specifically enzyme replacement treatment (ERT), also influences the immune response. BNP: B-type natriuretic peptide; CCR2: monocyte chemoattractant protein-1 receptor; CD: cluster of differentiation; DCs: dendritic cells; DD: diastolic dysfunction; DN iNKT: double-negative invariant natural killer T cells; ERT: enzyme replacement treatment; FGF2: fibroblast growth factor 2; GM-CSF: granulocyte–macrophage-colony-stimulating factor; IL: interleukin; INF-γ: interferon gamma; iNKT: invariant natural killer T cells; LGE: late gadolinium enhancement; LVH: left-ventricular hypertrophy; MCP-1: Monocyte Chemoattractant Protein-1; MHC: major histocompatibility complex; MMP: metalloproteinase; MR-proANP: midregional pro-atrial natriuretic peptide; No ≠: no differences; TGF: transforming growth factor; TLR: Toll-like receptor; TNF: tumor necrosis factor; TNFR: tumor necrosis factor receptor. Image has been adapted from [50].

**Figure 4 cells-13-00654-f004:**
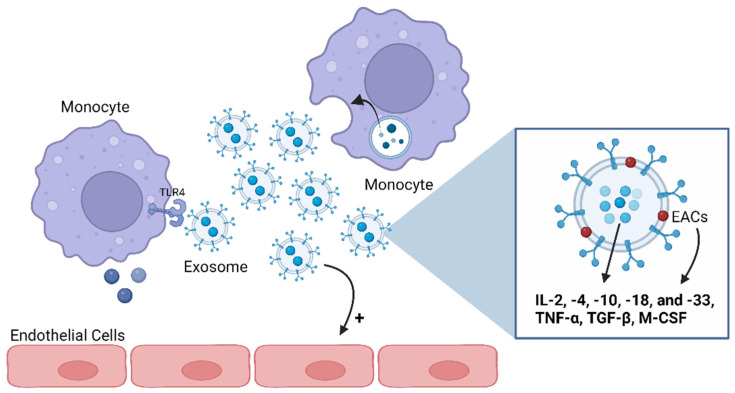
Exosomes’ functions as mediators of the immune and inflammatory pathways. Exosomes can carry anti- and pro-inflammatory cytokines both on their lumens and associated with their membranes. Several cytokines have been shown to be preferentially enriched within exosomes, including interleukin (IL)-2, -4, -10, -18, and -33; tumor necrosis factor (TNF)-α; transforming growth factor (TGF)-β; and macrophage-colony-stimulating factor (M-CSF). Thus, an immune cell can communicate with another through exosomes, namely by activating or inhibiting the production of cytokines in the target cell. Also, exosomes can stimulate endothelial cells. ECAs: exosome-associated cytokines; TLR4: Toll-like receptor 4. Image: created with biorender.com.

**Table 1 cells-13-00654-t001:** Differences in biogenesis, size, function, and cargo between the three groups of EVs (exosomes, MVs, and apoptotic bodies), adapted from [77].

	Exosomes	Microvesicles	Apoptotic Bodies
**Biogenesis**	Endocytic origin	Plasma membrane	Plasma membrane
**Size**	40–120 nm	50–1000 nm	500–2000 nm
**Function**	Intercellular communication	Intercellular communication	Facilitate phagocytosis
**Cargo**	Proteins and nucleic acids	Proteins and nucleic acids	Nuclear fractions, cell organelles

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
