# Peer review of "Inflammation and Exosomes in Fabry Disease Pathogenesis"

_cells, 2024, doi:10.3390/cells13080654_

Round 1
Reviewer 1 Report
Comments and Suggestions for Authors
An Overview of the Inflammatory Response and Future Perspectives on Fabry Disease
This is a review article about the inflammatory response in Fabry disease and, in parallel, the role of extracellular vesicles in this scenario. This is a great challenge, difficult to achieve and to transmit with clarity.
The article provides a very high amount of data, mainly experimental, regarding the mechanisms of chronic inflammation. However, there is big room for improvement in the way to present these data to facilitate comprehension. The graphs on the figures are very simple and don’t help much in understanding the pathophysiology of inflammation in Fabry disease.
The manuscript relies too much on a few published articles, some of them based on non-proved hypotheses. The extracellular vesicles and exosome section is mainly based on general knowledge with just 1-2 references on EVs and Fabry disease.
Thus, my view is that the manuscript needs a major revision or to be redone to facilitate the reading and improve the understanding of the inflammatory mechanisms in the pathophysiology of Fabry disease. In addition, at present, the future perspectives on Fabry disease must be much developed, beyond extracellular vesicles.
Lines 44-47: This is not well explained, it is not easy to understand. The message to transmit here is that opposite of what happens with other X-linked disorders, heterozygote females with Fabry disease mostly are symptomatic, although with a wide range of clinical expression. Rephrase it, please!
Lines 80-82: Please, describe better renal involvement in Fabry disease as a pathognomonic clinical feature: chronic kidney disease with decreased glomerular filtration rate (GFR) and proteinuria, progressing to end-stage renal failure.
Line 84: classical instead of severe phenotype.
Line 86: references 14, 15, 16, and 17 represent excessive auto-citation.
Line 86-88: please, rephrase the differential manifestations in males and females with Fabry disease.
Lines 90-91: please, elaborate English language.
Lines 96-99: All this is very speculative without a previous scientific basis.
Concerning the section 2. Cellular Mechanisms Involved in Fabry Disease Pathogenesis, many of the hypotheses drawn here are based on one article (reference # 27).
Lines 269-271: Is this telling us that the inflammatory response is not directly correlated to Gb3/lyso-Gb3 levels?
Line 272: Contribution of CKD with renal insufficiency?
Lines 291-292: Please, revise the references #35: Mauhin, #44: Rosa Neto
Lines 318-320: Altogether, these data suggested that clearance of Gb-3 may lead to suppressed immune responses, counteracting exaggerated autoimmune responses [48]. This is in contrast with “which suggests that ERT is not able to reduce the established systemic inflammatory state [43]”. Can you comment?
Line 358: cyclooxygenase-2 (COX-2)
Section 4. Extracellular Vesicles: A New Perspective in Fabry Disease. It isn't easy to see the adequacy of combining this section with the others in the same manuscript.
Author Response
This is a review article about the inflammatory response in Fabry disease and, in parallel, the role of extracellular vesicles in this scenario. This is a great challenge, difficult to achieve and to transmit with clarity.
The article provides a very high amount of data, mainly experimental, regarding the mechanisms of chronic inflammation. However, there is big room for improvement in the way to present these data to facilitate comprehension. The graphs on the figures are very simple and don’t help much in understanding the pathophysiology of inflammation in Fabry disease.
We thank Reviewer for this suggestion. For a better comprehension and reading, we restructured section 3 (Role of Inflammation in Fabry Disease: A Friend or a Foe?) by creating three subsections (3.1. Inflammatory Soluble Factors Profile in Fabry Disease; 3.2. Inflammatory Cells Profile in Fabry Disease; and 3.3. Influence of FD Treatment on Inflammatory Response). We also eliminated figure 4 and modified figure 1 and figure 3 to be more helpful for the understanding of the text.
The manuscript relies too much on a few published articles, some of them based on non-proved hypotheses. The extracellular vesicles and exosome section is mainly based on general knowledge with just 1-2 references on EVs and Fabry disease.
We thank Reviewer for this suggestion. We have added more references throughout the manuscript and when non-proved hypotheses were raised, the language was moderated to better transmit the readers the sense of conjecture.
Additionally, Section 4 (Extracellular Vesicles in Fabry Disease) is now based on seven references of eight that are so far, to the extent of our knowledge, the only ones available.
Thus, my view is that the manuscript needs a major revision or to be redone to facilitate the reading and improve the understanding of the inflammatory mechanisms in the pathophysiology of Fabry disease.
Authors agree with the Reviewer’s comment. In this context, a major revision of the manuscript and figures was done. Section 3 (Role of Inflammation in Fabry Disease: A Friend or a Foe?) was remodulated by subdividing it in three subsections (3.1. Inflammatory Soluble Factors Profile in Fabry Disease; 3.2. Inflammatory Cells Profile in Fabry Disease; and 3.3. Influence of FD Treatment on Inflammatory Response). Additionally, the title of section 4 was altered (Extracellular Vesicles in Fabry Disease) and information was added to enrich the manuscript. The visual support of the paper was also improved by eliminating figure 4 and altering figure 1 and figure 3. On section 2 (Cellular Mechanisms Involved in Fabry Disease Pathogenesis), new references were added, and an excerpt of section 3 was moved to section 2.
In addition, at present, the future perspectives on Fabry disease must be much developed, beyond extracellular vesicles.
We thank the Reviewer for pointing out this aspect on FD therapeutics. Nonetheless, we believe that a new section on FD therapeutics is out of scope of our review. To avoid this conflict, we altered the title of the manuscript (Inflammation and Exosomes on Fabry Disease Pathogenesis), the title of the section 4 (Extracellular Vesicles in Fabry Disease) and added information to the section 4. Additionally, we re-wrote some phrases to better outline the aim of our review:
Line 515-520: “In such manner, exosomes, due to their cargo and surface receptors, can play a range of functions on diverse processes, such as balance of immune responses and regulation of central and peripheral immunity, apoptosis, angiogenesis and wound healing, and host-microbiome interaction and viral immunity [84]. Therefore, exosomes may have an important role in the progression of FD through the modulation of the inflammatory cascades known to occur in this disorder.”
Line 582-586: “Adding to the equation the proven potential of exosomes to play a role as biomarkers in the diagnosis and prognosis of several diseases [92], the utility of these vesicles may be further expanded to FD diagnosis, prognosis, and early detection with cautionary note for their use in heterozygous females given their heterogenous disease phenotype.”
Line 596-599: “Altogether and considering the potential of exosomes as intermediators of inflammatory responses in several disorders, the utility of these EVs in the management of FD patients should be further explored, bringing new perspectives to diagnosis, prognosis, and early detection.”
Lines 44-47: This is not well explained, it is not easy to understand. The message to transmit here is that opposite of what happens with other X-linked disorders, heterozygote females with Fabry disease mostly are symptomatic, although with a wide range of clinical expression. Rephrase it, please!
We thank Reviewer for this suggestion. We believe we have clarified this matter. We copied below the changes made:
Line 40-45: “Females can exhibit a wide range of clinical manifestations, spanning from a lifelong absence of symptoms to experiencing a phenotype as severe as the one observed in males. In contrast to the pattern seen in many other X-linked disorders, heterozygous females with FD are predominantly asymptomatic. This variability in symptom presentation can be attributed to the random inactivation of one X-chromosome in each somatic cell during embryonic development, a process known as lyonization [4, 10]”
Lines 80-82: Please, describe better renal involvement in Fabry disease as a pathognomonic clinical feature: chronic kidney disease with decreased glomerular filtration rate (GFR) and proteinuria, progressing to end-stage renal failure.
We believe we have clarified this matter. We copied below the changes made:
Line 81-85: “Cardiac involvement is characterized by left ventricular hypertrophy (LVH), heart failure, and dysrhythmias. Renal manifestations typically initiate with tubular isosthenuria and hyperfiltration, followed by proteinuria and a decline in glomerular filtration rate (GFR) that may lead to end-stage renal failure. Cerebrovascular complications englobe strokes, transient ischemic attacks, and brain white matter lesions [14-17].”
Line 84: classical instead of severe phenotype.
We thank Reviewer for the suggestion. We followed it.
Line 86: references 14, 15, 16, and 17 represent excessive auto-citation.
We thank Reviewer for pointing out this issue. We rectified this excessive auto-citation.
Line 86-88: please, rephrase the differential manifestations in males and females with Fabry disease.
We thank Reviewer for this suggestion. We believe we have clarified this matter. We copied below the changes made:
Line 90-96: “For instance, LVH is observed in approximately 50% of affected males and 33% of FD females [17]. While in females, fibrosis commonly occurs without evidence of hyper-trophy, in males, LVH generally precedes fibrosis and ventricular dysfunction [17]. Regarding renal complications, chronic kidney disease (CKD) is more frequent in males presenting the FD classical phenotype, while proteinuria stands out as the most clinical prognostic marker for renal disease progression in both genders [17].”
Lines 90-91: please, elaborate English language.
We thank Reviewer for the suggestion. We copied below the changes made:
Line 97-98: “In Fabry Disease, the Gb-3 lysosomal deposition triggers pathogenic cascades that lead to chronic inflammation ending in cellular and organ damage [1].”
Lines 96-99: All this is very speculative without a previous scientific basis.
We fully agree with Reviewer, and we acknowledge the comment. We reinforced that such idea is a hypothesis that we truly believe could revolutionize FD management. We copied below the changes made, underlying the reinforcement of the conjecture:
Line 106-109: “We truly believe that the comprehension of the immune response and the involvement of EVs in FD could greatly benefit the identification of immunological pathways that would lead to the development of targeted therapeutics, and to advancements in its diagnosis and prognosis.”
Concerning the section 2. Cellular Mechanisms Involved in Fabry Disease Pathogenesis, many of the hypotheses drawn here are based on one article (reference # 27).
We thank the Reviewer for the comment. We revised the literature and enriched this section with more information regarding cellular mechanisms involved in Fabry disease pathogenesis, namely disrupted autophagy, and apoptosis. In addition, we moved a part that was written in section 3 to this section (Section 2) and we also reinforced that immune response in FD will be explored in detail in the next section (Section 3 - Role of Inflammation in Fabry Disease: A Friend or a Foe?).
Lines 269-271: Is this telling us that the inflammatory response is not directly correlated to Gb3/lyso-Gb3 levels?
We thank the Reviewer for the question. In line 269-271, we could not find the question related to Gb3/lyso-Gb3 levels. Nonetheless, given the relevance of the question and assuming it was based on line 273-276 from the original version submitted, we decided to add a new subsection (3.3. Influence of FD Treatment on Inflammatory Response) on section 3, exploring the effect of ERT on FD immune response. We also underlined a study that compares lyso-Gb3 levels in FD patients under ERT and FD patients without treatment. We pointed out that attention should be taken when analysing lyso-Gb3 levels on FD patients under ERT that are often not correlated with the action of ERT.
Line 382-386: “Contrasting, Yogasundaram et al. observed that ERT-treated FD patients revealed higher levels of TNF, TNFR1, TNFR2, MMP-2, and lyso-Gb3, compared to those not under ERT [57] (Figure 3). These results suggest that although plasma lyso-Gb3 is a potential primary screening biomarker for classical and late-onset FD [69], cautions should be taken when analyzing it in FD patients under ERT.”
Line 272: Contribution of CKD (chronical kidney disease) with renal insufficiency?
The authors struggled to understand Reviewers comment, but assuming that the Reviewer is referring to the line 309-312 in the new manuscript, the GFR levels were indicated for a better understanding of CKD contribution, as follows:
Line 311-314: “At last, patients with renal complications (patients with middle reduced kidney func-tion, estimated GFR of 60 mL/(min·1.73 m2)) showed increased plasma levels of BNP, MR-proANP, TNF-α, MMP-2, MMP-8, galectin-3, and galectin-1 [57].”
Lines 291-292: Please, revise the references #35: Mauhin, #44: Rosa Neto
We thank the Reviewer for the suggestion. We have rectified the references.
Lines 318-320: Altogether, these data suggested that clearance of Gb-3 may lead to suppressed immune responses, counteracting exaggerated autoimmune responses [48]. This is in contrast with “which suggests that ERT is not able to reduce the established systemic inflammatory state [43]”. Can you comment?
Authors do agree with Reviewer’s comment. In consequence of it, we added an entire subsection (3.3. Influence of FD Treatment on Inflammatory Response) focusing on the influence of ERT on immune response. We discussed hypotheses that may justify, in some cases, the persistency (or worsen) of the immune response on FD patients under ERT.
Line 358: cyclooxygenase-2 (COX-2)
We thank Reviewer for the correction.
Section 4. Extracellular Vesicles: A New Perspective in Fabry Disease. It isn't easy to see the adequacy of combining this section with the others in the same manuscript.
In order to better fit the EVs potential on FD, we changed the title of section 4 to “Extracellular Vesicles in Fabry Disease”. Since there are no references about EVs as inflammatory mediators on FD, we decided to leave some examples of EVs inflammatory role on other diseases, ending wondering about the existence of a parallelism in FD.

Reviewer 2 Report
Comments and Suggestions for Authors
This review takes into account the relation between Fabry disease (FD) and inflammation which is a frequent outcome of patients. The review is complete and well structured in relation to cellular mechanisms that are involved in inflammation and innate and adaptive immune-response. Extracellular vesicles represent an important and new challenge in the medical field. They may represent possible biomarkers for diagnosis, prognosis and therapy follow-up in many diseases, mostly cancer. Nevertheless it is difficult for me to understand (also from the manuscript) how they can be used as FD diagnostic tools, especially in the case of female heterozygosis. Can the Authors discuss these aspects in the manuscript?
As it is, Figure 1 doesn't add information to the manuscript. I suggest the Authors to change the figure in a schematic representation of Gb-3 and lyso-Gb-3 indicating where alpha-Gal-A is involved in the degradation of the molecules.
Author Response
Nevertheless it is difficult for me to understand (also from the manuscript) how extracellular vesicles can be used as FD diagnostic tools, especially in the case of female heterozygosis. Can the Authors discuss these aspects in the manuscript?
Authors acknowledged Reviewer for addressing this point. We think we have clarified this aspect in the paper by revising the entire section and further exploiting the EVs potential on FD. In addition, a cautionary note was added to highlight the concern regarding the application of EVs to heterozygous females. Below are the changes we have made:
Line 582-586: “Adding to the equation the proven potential of exosomes to play a role as biomarkers in the diagnosis and prognosis of several diseases [92], the utility of these vesicles may be further expanded to FD diagnosis, prognosis, and early detection with cautionary note for their use in heterozygous females given their heterogenous disease phenotype.”
As it is, Figure 1 doesn't add information to the manuscript. I suggest the Authors to change the figure in a schematic representation of Gb-3 and lyso-Gb-3 indicating where alpha-Gal-A is involved in the degradation of the molecules.
We altered Figure 1 according to the Reviewer suggestion (please see new Figure 1).

Reviewer 3 Report
Comments and Suggestions for Authors
This is a comprehensive review on the pathogenesis of Fabry disease and immune dysfunction and cascade pathways being triggered by Lyso GB3.
It would be helpful to quote studies where chaperone therapy is used in Fabry disease and what happens to these inflammatory markers.
Also to speculate why malignancy is not common with Fabry disease
Comments on the Quality of English LanguagePage 81 Please change conduct to lead to.
Author Response
It would be helpful to quote studies where chaperone therapy is used in Fabry disease and what happens to these inflammatory markers.
We thank Reviewer for addressing this point. To the extent of our knowledge, there are no research exploring the relation of immune response on FD patients receiving chaperone therapy, only with FD patients under ERT. We also pointed out this literature gap in the article.
Line 377-379: “To our current knowledge, there are no studies exploring the effect of chaperone therapy in FD immune response.”
Also to speculate why malignancy is not common with Fabry disease.
We thank Reviewer for the suggestion. We briefly addressed this topic on the paper. Below is the information added:
Line 98-102: “Curiously, FD patients seem to have a marginal reduction of cancers associated with elevated expression of Gb-3, while the stimulation by lyso-lipids, disease-related inflammation and vascular abnormalities could explain the increase incidence of melanoma urological malignancies and meningiomas [20].”

Round 2
Reviewer 1 Report
Comments and Suggestions for Authors
Title. Inflammation and Exosomes in Fabry Disease Pathogenesis. I suggest: The Role of Inflammation and Exosomes on Fabry Disease Pathogenesis.
I acknowledge the effort of the authors to improve the readability of the manuscript taking into account the high volume and the complexity of the information.
Nevertheless, I still have some minor comments:
- Page 1, lines 43-44: heterozygous females with FD are predominantly asymptomatic. The correct word is *symptomatic*.
- Figure 1: If authors pay attention to the referred original image, you will see that the lysosomes in the Fabry stage are “swollen” and increased in number. You should avoid including Lyso-Gb3 with the normal agal-A functioning and revise the arrow of deacetylase that should indicate the conversion of Gb-3 to Lyso-Gb3 in the Fabry stage. Please, modify.
- Page 2, line 84: followed by proteinuria. Please, change to microalbuminuria/proteinuria.
- Page 3, lines 99-103: I recommend eliminating this sentence that I think is out of context.
- Page 3, line 107: We truly believe. Eliminate “truly”.
- Page 5, line 174: Another related-Gb-3 substrate. Eliminate these words and start the phrase with Globotriaosylsphingosine.
- Page 6, line 251: the impact of FD treatment on the immune response will also be explored. Please, change “explored” to analysed, or described.
- Figure 3: Explain “no ¹” in the footnotes.
- Page 8, line 313: Regarding males, FD patients had significantly increases serum levels of IL-6 and TNF-α than male controls [55]. Change to: FD male patients had significantly increased serum levels of IL-6 and TNF-α compared to control males.
- Page 8, line 321: (patients with middle reduced kidney function, estimated GFR of 60 mL/(min·1.73 m2)) showed. Please, change “middle” to mildly or moderately reduced renal function, and estimated GFR <60 mL.
- Page 9, lines 384-385: improve FD patients’ quality of life. You should add… and protect from organ dysfunction increasing survival.
- Page 9, lines 389-390: eliminate lines 389 and 390, and change to “enhancing residual enzyme activity in patients with amenable mutations”.
- Page 9, lines 395-397: Please, be careful how you express this assertion, and include or modify in accord to: “These findings highlight the fact that patients who qualify for and receive ERT typically have more severe disease manifestations”.
- Concerning the bibliography, please revise all the references because there are many of them without the pages or the article number.
Author Response
Response to Reviewer 1
Title. Inflammation and Exosomes in Fabry Disease Pathogenesis. I suggest: The Role of Inflammation and Exosomes on Fabry Disease Pathogenesis.
We thank Reviewer for the correction.
- Page 1, lines 43-44: heterozygous females with FD are predominantly asymptomatic. The correct word is *symptomatic*.
- Figure 1: If authors pay attention to the referred original image, you will see that the lysosomes in the Fabry stage are “swollen” and increased in number. You should avoid including Lyso-Gb3 with the normal agal-A functioning and revise the arrow of deacetylase that should indicate the conversion of Gb-3 to Lyso-Gb3 in the Fabry stage. Please, modify.
- Page 2, line 84: followed by proteinuria. Please, change to microalbuminuria/proteinuria.
- Page 3, lines 99-103: I recommend eliminating this sentence that I think is out of context.
- Page 3, line 107: We truly believe. Eliminate “truly”.
- Page 5, line 174: Another related-Gb-3 substrate. Eliminate these words and start the phrase with Globotriaosylsphingosine.
- Page 6, line 251: the impact of FD treatment on the immune response will also be explored. Please, change “explored” to analysed, or described.
- Figure 3: Explain “no ¹” in the footnotes.
- Page 8, line 313: Regarding males, FD patients had significantly increases serum levels of IL-6 and TNF-α than male controls [55]. Change to: FD male patients had significantly increased serum levels of IL-6 and TNF-α compared to control males.
- Page 8, line 321: (patients with middle reduced kidney function, estimated GFR of 60 mL/(min·1.73 m2)) showed. Please, change “middle” to mildly or moderately reduced renal function, and estimated GFR <60 mL.
- Page 9, lines 384-385: improve FD patients’ quality of life. You should add… and protect from organ dysfunction increasing survival.
- Page 9, lines 389-390: eliminate lines 389 and 390, and change to “enhancing residual enzyme activity in patients with amenable mutations”.
- Page 9, lines 395-397: Please, be careful how you express this assertion, and include or modify in accord to: “These findings highlight the fact that patients who qualify for and receive ERT typically have more severe disease manifestations”.
- Concerning the bibliography, please revise all the references because there are many of them without the pages or the article number.
We apologize for the errors and we are thankful to Reviewer for the corrections. All of them were taken into account and modified accordingly to Reviewer’s comments.
